# Are the Benefits of a High-Emission Vehicle Driving Area Restriction Policy Greater than the Costs?

**DOI:** 10.3390/ijerph192315789

**Published:** 2022-11-27

**Authors:** Jia Zhou, Hongqiang Jiang, Xi Cheng, Yaling Lu, Wei Zhang, Zhanfeng Dong

**Affiliations:** 1The Center of Enterprise Green Governance, Chinese Academy for Environmental Planning, Beijing 100012, China; 2State Environmental Protection Key Laboratory of Environmental Planning and Policy Simulation, Chinese Academy for Environmental Planning, Beijing 100012, China

**Keywords:** cost–benefit analysis, high-emission vehicle, vehicle driving area restriction policy, Beijing–Tianjin–Hebei region, restriction, scrappage

## Abstract

China implemented a vehicle driving area restriction policy to control air pollution by delimiting a no-driving area for high-emission vehicles (HEVs). Current academic research does not evaluate the benefits and costs of this policy based on vehicle level and lacks evidence at the regional level. Therefore, a cost–benefit analysis method is developed to evaluate the benefits and the costs of this policy, based on 2.128 million HEVs in the Beijing-Tianjin-Hebei (BTH) region from 2008 to 2015. The benefits, the costs, and net benefits of this policy were CNY 98.49, CNY 5.80 and CNY 92.69 billion. The cost–benefit ratios of the BTH region, Beijing, Tianjin and Hebei were 1:16.98, 1:20.88, 1:14.52 and 1:16.55, respectively. Beijing’s cost–benefit ratio was the maximum at the city scale. This work can provide scientific support for promoting driving area restriction policies on older gasoline vehicles and older diesel vehicles, the improvement of cost–benefit analysis and efficient decision-making for the Chinese government.

## 1. Introduction

Vehicle emissions are currently a major source of air pollution. Gasoline vehicles and diesel vehicles emit a large amount of PM, NO_x_, CO and HC [1]. From 1990 to 2021, the quantity of China’s vehicles increased drastically from 5 million to 395 million. According to Beijing’s PM_2.5_ source apportionment results in 2021, the contribution from vehicle sources accounted for 46% (the largest among all pollution source types), which was higher than in 2017 (45%) and 1.45 times higher than in 2012 (31.1%) [2,3,4]. High-emission vehicles (HEVs) are usually yellow-label vehicles, which are gasoline vehicles that do not achieve the I emission standard of China and diesel vehicles that do not achieve the III emission standard of China. In 2014, there were 9.842 million HEVs in China, which constituted only 6.8% of all the vehicles, but accounted for 74.6% of PM, 47.4% of NO_x_, 45.4% of CO and 49.1% of HC emissions [1]. In order to control the increasing vehicle emissions, the Chinese government started restricting the driving areas of high-emission vehicles, named the high-emission vehicle driving area restriction (HVDAR) policy. The HVDAR policy delimits no-driving areas to control air pollution. According to this policy, the government has accelerated the scrappage of high-emission vehicles by restricting their driving areas. Beijing has implemented this policy since 2008. From 2013 to 2017, the government designated and expanded the no-driving areas for HEVs in cities nationwide.

The impacts of this policy on the people and society are huge and complex. Some people doubt whether the HEVs driving area restriction policy is worth doing. A cost–benefit analysis (CBA) is a useful tool of policy analysis and decision-making, but rarely used for the evaluation of vehicle driving area restriction policies. The value of a CBA for HVDAR was previously identified [5,6] and is still used to promote decision-making [7] in the literature. Some CBA studies have been carried out regarding HVDAR [8]. Some scholars have studied other impacts of HVDAR policies, such as gasoline consumption, gasoline prices, income [9] and car travel [10]. Moreover, some scholars only carried out parts of a CBA, such as the costs, cost efficiencies [11,12,13,14,15], emission reductions or the resulting air quality improvements of HVDAR [5,6,8,16,17]. Some overall studies have been carried out on the cost and benefit of the *Action Plan for Prevention and Control of Air Pollution* in China [18,19,20], which was issued by the State Council to control air pollution from 2013 to 2017.

The cost–benefit analysis of the restriction of high-emission vehicles was based on rough, restricted vehicle statistical data. Meanwhile, small-scale studies usually have incomplete, inexact and unquantifiable problems, as well as subsequent uncertainties of the CBA of HVDAR that could cause significant differences or even negligible results. The CBA served as a useful tool for decision-makers in identifying and quantifying the costs and benefits of the HVDAR. Moreover, CBAs are important for achieving various purposes, for example setting no-driving areas and driving restriction times, controlling the emission of pollutants from high-emission vehicles, and promoting and optimizing the development of the automotive industry. However, researchers cannot exactly assess the true value of high-emission vehicle driving restrictions in large-scale cases. As far as we know, no complete, exact and measurable CBAs on restricted high-emission vehicles exist on a large-scale. Thus, it is hard to know the effects of the restriction of HEVs and to recognize the true value of the HEV driving area restriction policy on a national or regional scale.

To answer this question, this paper constructs a CBA methodology to measure the costs and benefits of the HEV driving area restriction policy in the Beijing-Tianjin-Hebei (BTH) region, which is based on 2.128 million HEV observations. The results effectively demonstrate the true value of the HEV driving area restriction policy for a large-scale case. Moreover, the results can bring scientific and technological support for the implementation of driving area restriction policies for older gasoline vehicles and older diesel vehicles, the utilization and improvement of CBAs, and efficient decision-making for the Chinese government.

## 2. Materials and Methods

### 2.1. Study Area

The study was conducted in the BTH region, including Beijing, Tianjin and 11 cities (Shijiazhuang, Tangshan, Qinhuangdao, Handan, Xingtai, Baoding, Zhangjiakou, Chengde, Cangzhou, Langfang and Hengshui) in the Hebei Province. Due to 3 reasons, 13 cities were chosen as the study area. First, Beijing is a capital city. The emissions of air pollutants in the BTH region were some of the largest quantities and, in terms of environmental quality, is one of the worst in China. Second, the BTH region was the first area to conduct a HEV driving area restriction policy, in 2008. Since 2013, the HEV driving area restriction policy has come into operation on a national-scale. Third, the data for HEVs, policy start times, restricted driving areas, GDP, premature deaths, urban populations, death rates, numbers of inpatients and other data are easier to acquire than elsewhere in China.

### 2.2. Study Scheme

The following scenario were set: (1) the quantity of real-world HEV driving area restrictions outside of the policy are set as the business as usual (BAU) scenario; (2) the current HEV driving area restrictions within the policy are set the scenario 1, which include BAU scenario results. The difference between the BAU scenario and scenario 1 is the net quantity of restricted HEVs within this policy. Based on the net quantity of restricted HEVs, the costs and benefits of the HEV driving area restriction policy are estimated. Table 1 analyzes the influence matrix of the HEV driving area restriction policy for different groups.

In Figure 1, first, the net quantity of restricted HEVs within the policy is estimated. Second, the direct costs are estimated from the restriction area and number of HEVs. Third, the emission reductions of restricted HEVs are estimated. Fourth, the Weather Research Forecasting–Community Multiscale Air Quality (WRF–CMAQ) system is used to simulate the impact of PM_2.5_ on ambient concentrations from 2008 to 2015. Fifth, the concentration reduction of PM_2.5_ is estimated through concentration–response functions and economic valuation models. Sixth, the costs, benefits and net benefits are compared.

### 2.3. Cost Analysis Method

For society, the costs are the expenses related to changing to other means of transportation. These costs are calculated according to the quantity of restricted HEVs, the average occupancy of vehicles, and the expense of changing to other means of transportation. The formula is as follows.
(1)$$C_{d}=\theta \times V_{d}\times P_{b}\times M$$
where *C_d_* is the expense of changing to other means of transportation (in CNY); *θ* is the average occupancy of the vehicle (*person per vehicle*); *V_d_* is the quantity of restricted HEVs; *P_b_* is the average annual expense on public transportation per person (CNY *per person per* km); and *M* is the average annual mileage of vehicles (km).

### 2.4. Benefit Analysis Methods

For society, the benefits include the HEV restriction-related transportation savings and health benefits. The HEV restriction-related transportation savings are calculated by the quantity of restricted HEVs, the average occupancy of vehicles, the expenses of car travel, and the average annual mileage of vehicles. The health benefits are calculated through emissions, air quality, health effect estimation and economic assessment.

#### 2.4.1. HEVs Restriction-Related Transportation Savings

The formula for HEV restriction-related transportation savings (*C_dc_*) is as follows.
(2)$$C_{dc}=\theta \times V_{d}\times P_{c}\times M$$
where *C_dc_* are the HEV restriction-related transportation savings (CNY); *θ* is the average occupancy of vehicles (*person per vehicle*); *V_d_* is the quantity of restricted HEVs; *P_c_* is the average annual expense of car travel per person (CNY *per person per* km); and *M* is the average annual mileage of vehicles (km).

#### 2.4.2. Health Benefits

##### Emissions Model

The PM_2.5_ emission reduction is calculated according to the Technical Guidelines for On-road Vehicle Air Pollutant Emission Inventory Compilation [21] and the Method for the Estimation of Vehicular Air Pollutant Emission in Urban Areas (HJ/T 180-2005) [22]. The formula is as follows.
(3)$$E=V\times EF\times M\times 10^{-6}$$
where *E* is the annual PM_2.5_ emission of HEVs (*t*), *V* is the quantity of restricted HEVs, *EF* is the quantity of pollutants emitted by HEVs (*gram per* km) and *M* is the average annual mileage of vehicles (km).

##### Air Quality Model

The WRF-CMAQ was used to quantify the influences of the policy on PM_2.5_ concentrations from 2008 to 2015. The initial boundary conditions were acquired from Final Operational Global Analysis 1° × 1° data of the National Centers for Environmental Prediction. The model is developed for domain 1 (resolution of 27 km), domain 2 (resolution of 9 km), and domain 3 (resolution of 3 km). Domain 1 covers the entire East China region, domain 2 covers North China and domain 3 covers the BTH region and some surrounding provinces. The model was run for January, April, July and October as representative months for spring, summer, autumn and winter, respectively. Boundary conditions were established based on the outputs of coarser grids. The number of vertical layers is 9. The gas phase chemical mechanism is the C0B5 scheme.

The emissions inventory consists of two types of emissions data. Firstly, the background emissions data acquired from the Multi-resolution Emission Inventory for China (MEIC v2012). Secondly, the annual pollutant emissions of restricted HEVs, which are included in the model. The concentration increments after superimposing vehicle emissions to the background emissions are the PM_2.5_ concentration contribution of the policy.

Monitoring data of PM_2.5_ in January, April, July and October from 45 national fixed air quality monitoring stations were acquired from the National Urban Air Quality Real-time Release Platform of the China National Environmental Monitoring Centre, which are used to evaluate the validity and accuracy of WRF-CMAQ simulation results. The correlation coefficient, mean fractional bias, and mean fractional error for statistical indices are used to validate and correct the results.

##### Health Effects Estimation

PM_2.5_ is a major air pollutant affecting human health [23,24]. Therefore, PM_2.5_ was selected as the pollutant factor to assess health effects. Chronic mortality, acute mortality, respiratory disease, cardiovascular disease and chronic bronchitis were selected as the health endpoints of PM_2.5_. In this study, the exposure–response function was utilized to evaluate the coefficient of correlation (*β*) between PM_2.5_ concentration and five health endpoints of the exposed population [25,26], including chronic mortality (0.00296), acute mortality (0.0004), respiratory disease (0.00109), cardiovascular disease (0.00068) and chronic bronchitis (0.01009). The exposure–response function is as follows.
(4)$$RR=\left[\left(C+1\right)/\left(C_{0}+1\right)\right]\beta$$
where *C* is the present concentration of PM_2.5_, *C*_0_ is the baseline concentration of PM_2.5_, *RR* is the relative risk of human health effect of PM_2.5_ and *β* is the exposure–response coefficient, indicating the percent (%) increase in the health endpoint for each unit of increase in PM_2.5_ concentration.

##### Economic Valuation

Economic loss caused by chronic mortality and acute mortality (*EL*_1_) were calculated by the human capital method. Economic loss (*EL*_2_) caused by respiratory disease and cardiovascular disease were calculated by the disease cost method. Economic loss *(EL*_3_) caused by chronic bronchitis were calculated in the disability-adjusted life years method [27]. A discount rate of 8% was chosen in the monetized benefits estimation, according to the *China Statistical Yearbook*. The formula of health benefits ($EL_{\mathrm{Total}}$) is as follows.
(5)$$EL_{\mathrm{Total}}=EL_{1}+EL_{2}+EL_{3}$$

### 2.5. Data

Restricted HEV data (e.g., policy start times, restricted driving areas, quantity and type of HEV) and other data (e.g., GDP, urban populations, premature deaths, death rates and numbers of inpatients) in Beijing, Tianjin, Shijiazhuang, Tangshan, Qinhuangdao, Handan, Xingtai, Baoding, Zhangjiakou, Chengde, Cangzhou, Langfang and Hengshui were collected from various sources, including the annual local statistical bulletin, annual local government work report, the website of the Ministry of Ecology and Environment, the *China Statistical Yearbook 2009–2016, the Beijing Statistical Yearbook 2009–2016, the Tianjin Statistical Yearbook 2013–2016*, the *Hebei Economic Yearbook 2014–2016* and *the China Health and Family Planning Yearbook 2015* [28]. The HEV driving area restriction policy took effect in Beijing from 2008 to 2015, in Tianjin from 2012 to 2015 and in the Hebei Province from 2013 to 2015.

## 3. Results

### 3.1. Quantity of Restricted HEVs

As shown in Figure 2, 402,000, 530,000, 195,000, 174,000, 63,000, 103,000, 84,000, 175,000, 60,000, 36,000, 139,000, 111,000 and 56,000 HEVs were restricted in Beijing, Tianjin, Shijiazhuang, Tangshan, Qinhuangdao, Handan, Xingtai, Baoding, Zhangjiakou, Chengde, Cangzhou, Langfang and Hengshui, respectively. There were 2.128 million restricted HEVs in the BTH region. The Hebei Province had 1.196 million restricted HEVs. The quantity of HEVs in Beijing, Tianjin and the Hebei Province accounted for 19%, 25% and 56% of all vehicles, respectively. Tianjin restricted the largest quantity of HEVs (530,000) at the city level, whereas Chengde restricted the lowest quantity of HEVs (36,000). Shijiazhuang restricted the highest quantity of HEVs (195,000) in the Hebei Province. Beijing implemented this policy from 2008 to 2015, and the driving of HEVs was not permitted within the sixth ring road of Beijing during this implementation time. Tianjin and the Hebei Province phased out HEVs with driving area restrictions from 2012 to 2015 and from 2013 to 2015. All the cost evaluations and benefit evaluations in this paper were based on 2.128 million HEVs.

### 3.2. Costs

Figure 3 shows that the costs of the driving area restriction policy in Beijing, Tianjin, Shijiazhuang, Tangshan, Qinhuangdao, Handan, Xingtai, Baoding, Zhangjiakou, Chengde, Cangzhou, Langfang and Hengshui were 1.36 billion RMB (i.e., CNY), CNY 1.69 billion, CNY 0.46 billion, CNY 0.42 billion, CNY 0.15 billion, CNY 0.23 billion, CNY 0.18 billion, CNY 0.40 billion, CNY 0.13 billion, CNY 0.08 billion, CNY 0.32 billion, CNY 0.24 billion and CNY 0.14 billion, respectively. Based on 2.128 million HEVs, the cost of the driving area restriction policy in the BTH region was CNY 5.80 billion. The cost in the Hebei Province was CNY 2.75 billion. The costs in Beijing, Tianjin and the Hebei Province accounted for 23%, 29% and 48% of the total costs, respectively. Tianjin’s cost was the highest (CNY 1.69 billion) at the city level, and Chengde’s cost was the lowest (CNY 0.08 billion). Shijiazhuang had the highest level of cost (CNY 0.46 billion) in the Hebei Province. Because of the differences in start year, completeness and restriction areas of this policy, the costs in each city were different. For example, the policy’s start year of Beijing and Tianjin were 2008 and 2012, respectively, whereas Chengde’s was 2015.

### 3.3. Benefits

#### 3.3.1. HEVs Restriction-Related Transportation Savings

In Figure 4, the HEVs restriction-related transportation savings of Beijing, Tianjin, Shijiazhuang, Tangshan, Qinhuangdao, Handan, Xingtai, Baoding, Zhangjiakou, Chengde, Cangzhou, Langfang and Hengshui were CNY 14.94 billion, CNY 18.68 billion, CNY 6.12 billion, CNY 5.61 billion, CNY 1.95 billion, CNY 3.10 billion, CNY 2.44 billion, CNY 5.30 billion, CNY 1.78 billion, CNY 1.09 billion, CNY 4.21 billion, CNY 3.18 billion and CNY 1.84 billion, respectively. The HEVs restriction-related transportation savings of the BTH region reached CNY 70.24 billion from 2008 to 2015. HEVs restriction-related transportation savings in the Hebei Province was CNY 36.62 billion. HEVs restriction-related transportation savings of Beijing, Tianjin and Hebei Province accounted for 21%, 27% and 52% of the total savings, respectively. Tianjin’s restriction-related transportation savings were the highest (CNY18.68 billion) at the city level, and Chengde’s restriction-related transportation savings were the lowest (CNY 1.09 billion). Shijiazhuang had the highest restriction-related transportation savings (CNY 6.12 billion) in the Hebei Province. Similar to the cost analysis of the HEVs driving area restriction policy, the savings analysis was also affected by start year, completeness and restriction areas of this policy.

#### 3.3.2. Health Benefits

Due to the implementation of the HEV driving area restriction policy, based on 2.128 million restricted HEVs, the emission reductions of PM_2.5_, PM_10_, NO_x_, CO and HC were 10.96 kt, 12.08 kt, 116.52 kt, 438.39 kt and 60.46 kt, in the BTH region from 2008 to 2015, respectively. As Figure 5a shows, PM_2.5_ emission reductions in Beijing, Tianjin, Shijiazhuang, Tangshan, Qinhuangdao, Handan, Xingtai, Baoding, Zhangjiakou, Chengde, Cangzhou, Langfang and Hengshui were 0.80 kt, 1.58 kt, 1.22 kt, 1.23 kt, 0.49 kt, 0.86 kt, 0.68 kt, 1.08 kt, 0.62 kt, 0.23 kt, 1.07 kt, 0.75 kt and 0.35 kt, respectively. The PM_2.5_ emission reduction in the Hebei Province was 8.58 kt. PM_2.5_ emission reductions in Beijing, Tianjin and the Hebei Province amounted for 7%, 15% and 78% of the total emissions, respectively. The PM_2.5_ emission reduction for restricted HEVs (10.96 kt) in the BTH region accounted for 6.32% of all vehicle PM_2.5_ emissions and 0.34% of all PM_2.5_ emissions in the BTH region. There was a positive correlation between vehicle type and quantity of HEVs and the reduction in pollutant emissions. Heavy HEVs used to emit more pollutants than light HEVs. Additionally, the greater the number of restricted HEVs, the more pollutant emissions could be reduced.

In Figure 5b, because of the implementation of the HEV driving area restriction policy, PM_2.5_ concentration reductions in Beijing, Tianjin, Shijiazhuang, Tangshan, Qinhuangdao, Handan, Xingtai, Baoding, Zhangjiakou, Chengde, Cangzhou, Langfang and Hengshui were 1.57 μg/m^3^, 1.42 μg/m^3^, 1.22 μg/m^3^, 1.52 μg/m^3^, 0.17 μg/m^3^, 0.35 μg/m^3^, 0.21 μg/m^3^, 1.3 μg/m^3^, 0.89 μg/m^3^, 0.12 μg/m^3^, 0.87 μg/m^3^, 0.60 μg/m^3^ and 0.68 μg/m^3^, respectively. PM_2.5_ concentration reductions in the Hebei Province were 7.93 μg/m^3^. PM_2.5_ concentration reductions in Beijing, Tianjin and the Hebei Province accounted for 14%, 13% and 73% of the total concentration reductions, respectively. The annual concentrations of PM_2.5_ in Beijing from 2008 to 2015, in Tianjin from 2012 to 2015 and in the 11 cities in the Hebei Province from 2013 to 2015 were reduced by 0.02–0.70 μg/m^3^, 0.00–0.79 μg/m^3^ and 0.00–1.07 μg/m^3^, respectively. The pollutant concentration reduction is obviously affected by the pollutant emission reduction, topography, geographical situation and meteorological conditions of every city. The maximum emission reduction in urban pollutants is not equal to the maximum reduction in pollutant concentration.

In Figure 5c, Beijing, Tianjin, Shijiazhuang, Tangshan, Qinhuangdao, Handan, Xingtai, Baoding, Zhangjiakou, Chengde, Cangzhou, Langfang and Hengshui had 2860, 1148, 828, 702, 32, 168, 81, 782, 252, 27, 349, 146 and 168 premature deaths (chronic premature deaths and acute premature deaths), respectively. The Hebei Province had 3535 premature deaths. The number of hospitalized people (respiratory diseases inpatients and cardiovascular diseases inpatients) in Beijing, Tianjin, Shijiazhuang, Tangshan, Qinhuangdao, Handan, Xingtai, Baoding, Zhangjiakou, Chengde, Cangzhou, Langfang and Hengshui were 33,482, 12,612, 8260, 6520, 309, 1759, 809, 8075, 2293, 261, 3550, 1533 and 1689, respectively. The number of hospitalized people in the Hebei Province was 35058. The number of chronic bronchitis patients in Beijing, Tianjin, Shijiazhuang, Tangshan, Qinhuangdao, Handan, Xingtai, Baoding, Zhangjiakou, Chengde, Cangzhou, Langfang and Hengshui were 8572, 3441, 2476, 2102, 97, 505, 242, 2342, 753, 83, 1046, 439 and 505, respectively. The number of chronic bronchitis patients in the Hebei Province was 10590. Because their driving restriction policy’s start year in Beijing was the earliest (2008) and Beijing was densely populated, Beijing had the largest number of people.

Figure 5d shows that the health benefits of the BTH region totaled CNY 28.25 billion, 93.88% of which come from premature deaths and chronic bronchitis. Health benefits of premature deaths, chronic bronchitis and hospitalizations were CNY 11.92 billion, CNY 14.62 billion and CNY 1.71 billion. Health benefits in Beijing, Tianjin, Shijiazhuang, Tangshan, Qinhuangdao, Handan, Xingtai, Baoding, Zhangjiakou, Chengde, Cangzhou, Langfang and Hengshui were CNY 13.36 billion, CNY 5.91 billion, CNY 2.21 billion, CNY 3.11 billion, CNY 0.07 billion, CNY 0.34 billion, CNY 0.11 billion, CNY 1.19 billion, CNY 0.42 billion, CNY 0.05 billion, CNY 0.87 billion, CNY 0.35 billion and CNY 0.26 billion, respectively. Health benefits were CNY 8.98 billion for the Hebei Province. Health benefits in Beijing, Tianjin and Hebei Province accounted for 47%, 21% and 32% of all total health benefits, respectively. Health benefits in Beijing, Tianjin and the Hebei Province (chronic premature deaths and acute premature deaths) were CNY 5.50 billion, CNY 2.52 billion and CNY 3.90 billion in premature deaths, CNY 1.12 billion, CNY 0.30 billion and CNY 0.29 billion in hospitalized people (respiratory diseases inpatients and cardiovascular diseases inpatients), and CNY 6.74 billion, CNY 3.09 billion and CNY 4.79 billion in chronic bronchitis inpatients, respectively.

### 3.4. Net Benefits

Figure 6 shows that the net benefits in the BTH region totaled CNY 92.69 billion, which means the benefits of the high-emission vehicle driving area restriction policy are far higher than the costs. The net benefits of Beijing, Tianjin, Shijiazhuang, Tangshan, Qinhuangdao, Handan, Xingtai, Baoding, Zhangjiakou, Chengde, Cangzhou, Langfang and Hengshui were CNY 26.94 billion, CNY 22.89 billion, CNY 8.29 billion, CNY 7.87 billion, CNY 3.29 billion, CNY 6.09 billion, CNY 2.07 billion, CNY 1.88 billion, CNY 1.07 billion, CNY 1.96 billion, CNY 3.22 billion, CNY 2.37 billion and CNY 4.75 billion, respectively. The net benefit in the Hebei Province was CNY 42.86 billion. The discount rate of net present value of the cost and benefit evaluations from 2008 to 2015 were 8%. The net benefits of all the 13 cities were greater than zero. Beijing’s net benefit was the highest, whereas Zhangjiakou’s net benefit was the lowest (CNY 1.07 billion) from 2013 to 2015 at the city level. And Shijiazhuang’s net benefit (CNY 8.29 billion) was greater than those of the other cities in the Hebei Province.

As Figure 6 shows, the net benefits were the highest for the first year of policy implementation. In Beijing, for example, the annual net benefits were CNY 7.06 billion, CNY 4.29 billion, CNY 2.82 billion, CNY 2.82 billion, CNY 2.74 billion, CNY 2.59 billion, CNY 2.36 billion and CNY 2.26 billion in 2008, 2009, 2010, 2011, 2012, 2013, 2014 and 2015, respectively. Moreover, the cost–benefit ratio for the BTH region was 1:16.97, which means a cost of CNY 1 would yield a benefit of CNY 16.97. The cost–benefit ratios for Beijing, Tianjin and the Hebei Province were 1:20.88, 1:14.52 and 1:16.55, respectively. Beijing’s cost–benefit ratio was consistently the maximum, and the cost–benefit ratio for Qinhuangdao (1:13.79) was the minimum at the city level.

In terms of the composition of net benefits, the vehicle restriction-related transportation savings were the largest part. The vehicle restriction-related transportation savings were determined by the quantity of restricted vehicles, average occupancy of vehicles, expenses of car travel, and average annual mileage of vehicles. When the average occupancy of vehicles, expenses of car travel, and average annual mileage of vehicles were fixed, the quantity of restricted vehicles directly determined the vehicle restriction-related transportation savings. For example, Beijing’s HEVs had the largest quantity of cars in the BTH region (382.8 thousand cars from 2008 to 2015). Therefore, the vehicle restriction-related transportation savings in Beijing were the highest, and the benefit in Beijing was relatively highest.

## 4. Discussion

Based on a total of 2.128 million HEVs from 2008 to 2015, region-level cost–benefit analysis methods were built under the disturbance of vehicular factors and environmental factors to evaluate the cost, benefit and net benefit. Based on vehicle-level observation data in real-world vehicular management, the results and subsequent policy recommendations can be easily transplanted into vehicular management and environmental management in China. The explorations of cost–benefit analysis method design can empirically support the prevention and control of air pollution from moving sources nationwide, as well as the policy formulation direction for tightening driving area restrictions.

This paper has some advantages over other papers. Firstly, bottom-up cost–benefit evaluation methods based on vehicle-level data are used. Therefore, the simulation results of cost–benefit evaluation are exact and reflect reality. Moreover, as all evaluation methods are built based on the observation data in vehicular management and environmental management in China, the method explores the empirical operability for future improvement in moving source management policy. On the other hand, previous papers mostly use top-down approaches to estimate the cost, cost-effectiveness or the benefit [14,15], hence it is difficult to afford opinions on improving region-level cost–benefit analysis methods based on vehicle-level observation data in China.

Secondly, the interference of vehicular factors and environmental factors, including vehicle type, fuel type, emissions factor and correction factors, are considered in our paper. Consequently, our built method successfully reduces the PM_2.5_ emissions of vehicles by 6.32% and PM_2.5_ emissions by 0.34% from 2008 to 2015 in the BTH region, such that the associated cost is only CNY 5.80 billion, the total benefits are CNY 98.49 billion, and the resulting net benefits are CNY 92.69 billion. The benefits of the HVDAR policy are far greater than the costs. Overall, the various implementations of the vehicle driving area restriction policy are advantageous in reducing the pollutant emissions of vehicles to build a cleaner environment.

Thirdly, by coupling the WRF-CMAQ model, the PM_2.5_ concentration reductions achieved with the vehicle driving area restriction policy can be quantified to establish relations between this policy and the air quality improvements. These results show that the policy based on vehicle-level data could reduce PM_2.5_ concentrations by 10.92 μg/m^3^, from 2008 to 2015. On the other hand, few previous vehicle-level papers analyze the influence of the policy on regional air quality because of inadequate vehicle-level observational data [5,6,17]. The assessments of emission reductions and air quality improvements are very advantageous for a complete and insightful understanding of the effects of this policy on the society and the environment.

This paper also has a few disadvantages. Firstly, because of data limitations, this paper does not consider all vehicles that are enforced in the driving area restriction policy, such as trucks and non-road vehicles. Fortunately, the practices of high-emission vehicles in this study cover 95.1%, 90.4% and 78.6% of the yellow-label vehicles in Beijing, Tianjin and the Hebei Province, and supply a good presentation for other vehicles to build cost–benefit analysis methods. Secondly, because of data limitations, this paper is a pilot study in the BTH region, and it is difficult to conduct a comprehensive analysis of high-emission vehicle driving area restriction policies on a national scale. After obtaining more information about high-emission vehicle driving area restriction policies, we could further study the cost–benefit analysis based on the proposed methods under various vehicle driving area restriction backgrounds. Thirdly, the selection of parameters in the cost and benefit analysis methods, the emissions inventory and meteorological data used in the air quality simulations may introduce uncertainty into the simulation results.

Several policy recommendations for the enforcement of the vehicle driving area restriction policy are given. Firstly, mandatory policy should be maintained and strengthened to promote the large-scale restriction and scrappage of older gasoline vehicles and older diesel vehicles, particularly in densely populated areas like big cities. An HEV driving area restriction policy would produce higher net benefits if enforced earlier. Moreover, the effect will be better in densely populated areas and regions with high GDP per capita. It is urgent and necessary to expand the restricted driving areas to restrict and scrap older gasoline vehicles and older diesel vehicles. A larger restricted driving area will lead to greater pollutant emission reductions and improved air quality. In future evaluations of the policy, all types of vehicles should be considered on a national scale.

Secondly, publicity, education and popularization should be further strengthened to encourage and promote car owners to use more renewable energy vehicles and public transport. Cities, especially big cities, should be encouraged to launch driving restriction area policies for older gasoline vehicles and older diesel vehicles as soon as possible.

Thirdly, in the current decision-making process for China’s environmental policy, it is necessary to further strengthen the understanding and application of cost–benefit analysis in China’s environmental policy in order to better evaluate an environmental policy that is comprehensive as well as systematic. Cost–benefit analysis is a very useful and powerful tool in supporting decision-making. However, there is still great space for promoting the application of cost–benefit analysis in the Chinese government.

Fourthly, a cost–benefit analysis system needs to be further improved, and the application of cost–benefit analysis needs to be further promoted in China’s environmental policy evaluation. Although there are some technical guidelines and specifications for cost–benefit analysis, it is not enough to form a cost–benefit analysis system. At present, it is still important and necessary to strengthen the legal system, as well as improve the technical methods and the specific case practice of cost–benefit analysis.

## 5. Conclusions

Based on actual data of 2.128 million HEV observations from 2008 to 2015, the HEV driving area restriction policy in the BTH region incurred the costs of CNY 5.80 billion and benefits of CNY 98.49 billion, created net benefits of CNY 92.69 billion. Net benefits were CNY 26.94 billion, CNY 22.89 billion and CNY 42.86 billion in Beijing, Tianjin and the Hebei Province, respectively. A case study of analyzing the costs and the benefits of the vehicle driving area restriction policy in the BTH region shows that implementing the policy could reduce the total PM_2.5_ emissions by 0.34% and reduce PM_2.5_ pollution concentration by 10.92 μg/m^3^ in the BTH region, based on data from 2008 to 2015. In addition, the benefits of the policy were much greater than the costs. It is necessary and worth continuing to promote the policy implementation to better control vehicular pollutant emissions. Therefore, the vehicle driving area restriction policy has played a significant role in reducing vehicular pollutant emissions and improving air quality of the city and the region. The evaluation method based on a cost–benefit analysis constructed in this paper can effectively and accurately evaluate the costs and benefits of the HVDAR policy. Additionally, the cost–benefit analysis method can support government decision-making.

## Figures and Tables

**Figure 1 ijerph-19-15789-f001:**
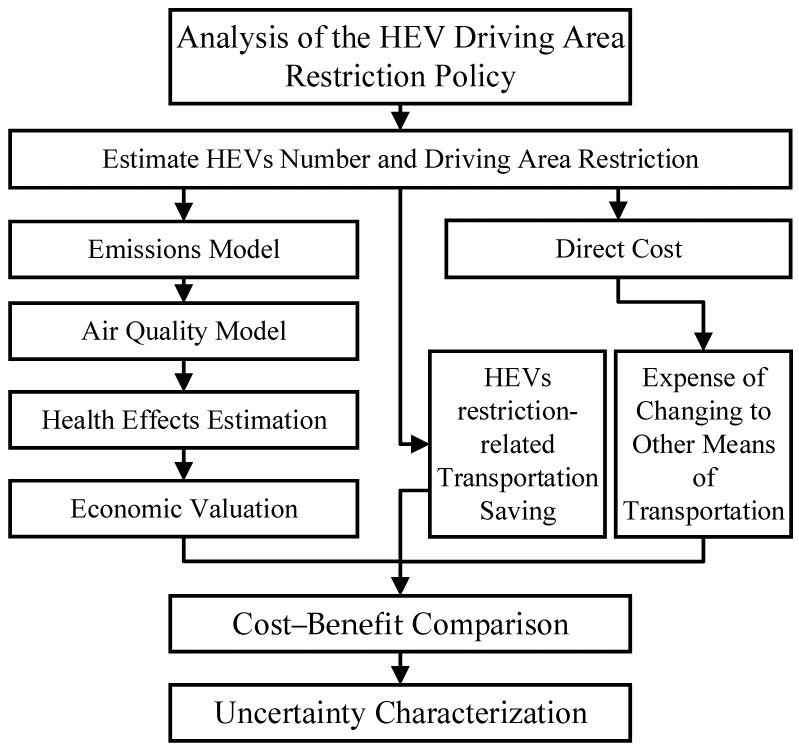
Analytical sequence of the HEV driving area restriction policy.

**Figure 2 ijerph-19-15789-f002:**
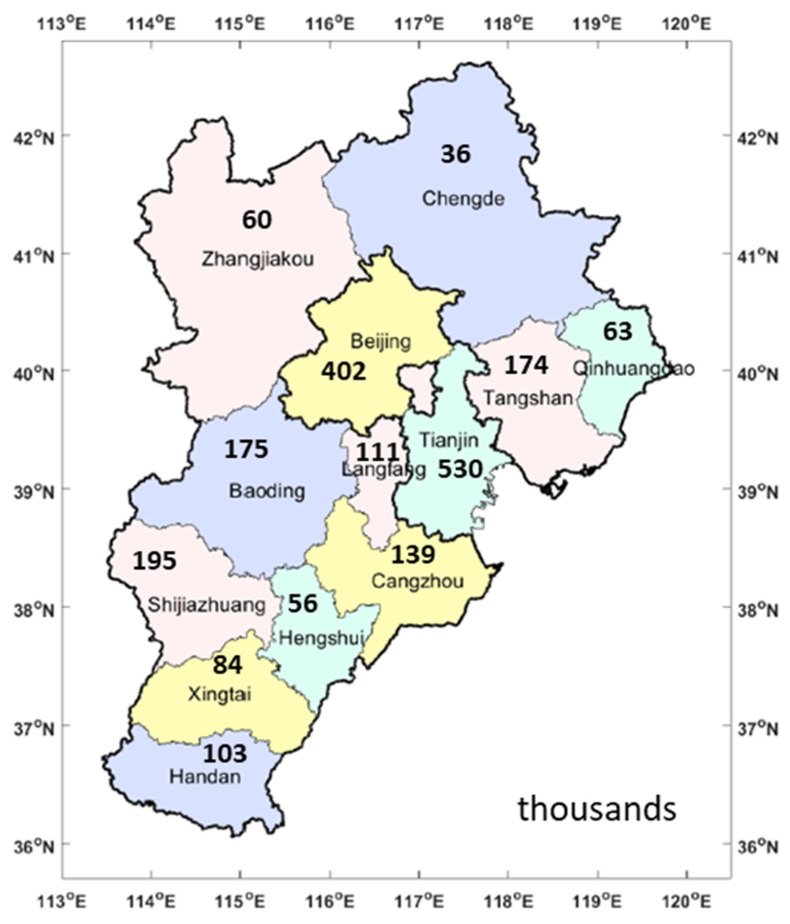
Quantity of restricted HEVs in the BTH region from 2008 to 2015.

**Figure 3 ijerph-19-15789-f003:**
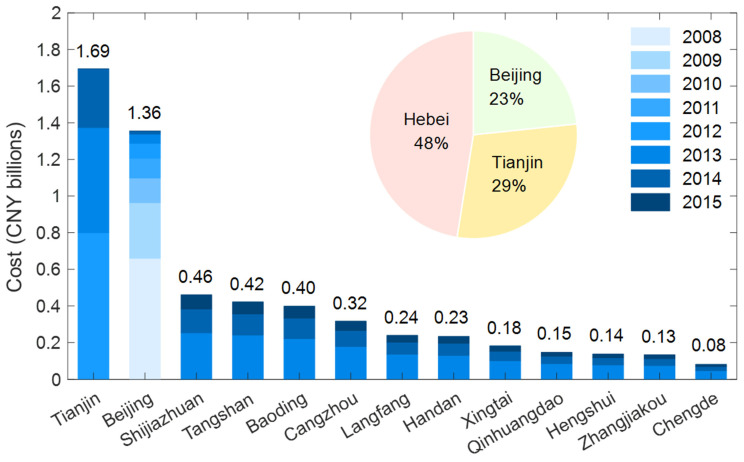
Cost of the HEV driving area restriction policy.

**Figure 4 ijerph-19-15789-f004:**
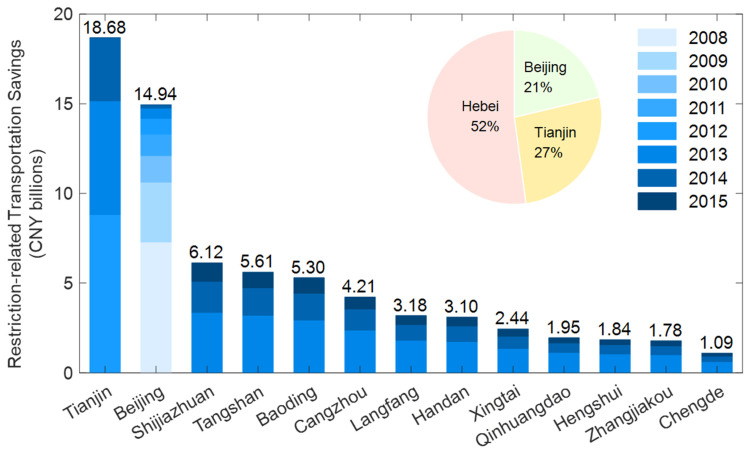
Restriction-related transportation savings from HEV driving area restriction policy.

**Figure 5 ijerph-19-15789-f005:**
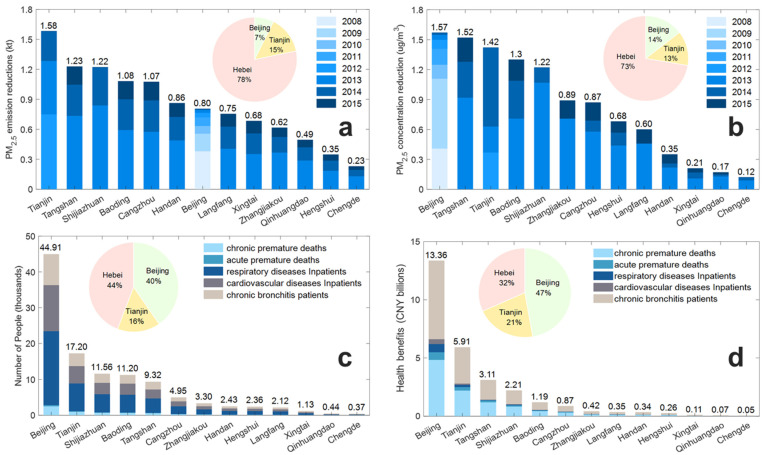
PM_2.5_ emission reductions (**a**), PM_2.5_ concentration reduction (**b**), number of people at the health endpoints (**c**), and health benefits (**d**) due to the implementation of the HEV driving area restriction policy.

**Figure 6 ijerph-19-15789-f006:**
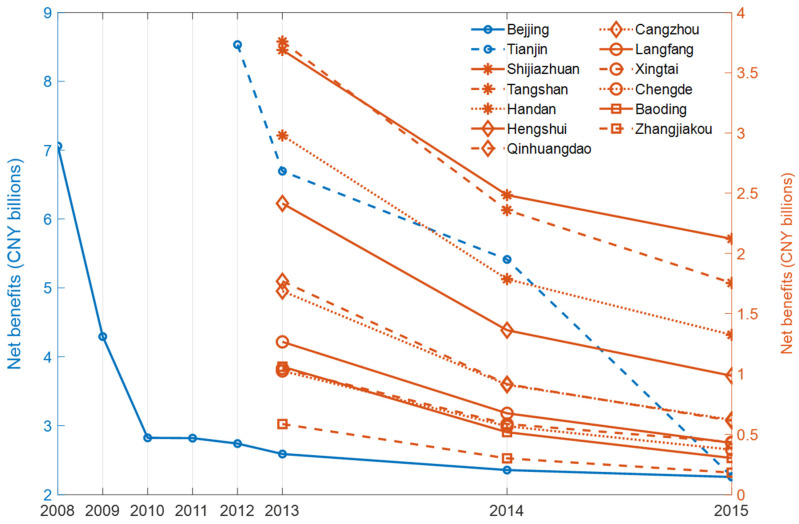
Net benefits of the HEV driving area restriction policy from 2008 to 2015.

**Table 1 ijerph-19-15789-t001:** Influence matrix of the HEV driving area restriction policy.

Group	Positive Influence	Negative Influence
Government	-	Management cost
Resident	Health (environmental quality) benefit; restriction-related transportation savings	Expense of changing to other means of transportation
Enterprise	-	-
Society	Health (environmental quality) benefit; restriction-related transportation savings	Management cost; expense of changing to other means of transportation

Note: Society includes the government, residents and enterprises. For society, all the costs and benefits of the government, residents and enterprises are offset, and health (environmental quality) benefits, restriction-related transportation savings, the management cost and expense of changing to other means of transportation are left. Management cost is too small to calculate, compared with the expense of changing to other means of transportation [7].

## Data Availability

The data are not publicly available, and any inquiries may be addressed by the first author.

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
