# Peer review of "Are the Benefits of a High-Emission Vehicle Driving Area Restriction Policy Greater than the Costs?"

_ijerph, 2022, doi:10.3390/ijerph192315789_

Round 1

Reviewer 1 Report

Dear Editor: Very glad to review this paper (ijerph-2031266). Thanks for your trust. This paper constructs a CBA methodology to quantify the cost and the benefit of high-emission vehicles driving area restriction policy in the Beijing-Tianjin-Hebei (BTH) region, which is based on actual data on 2.128 million HEV observations. The results can bring scientific and technological support for the implementation of driving area restriction policies for older gasoline vehicles and older diesel vehicles, the utilization and improvement of CBA, and efficient decision-making in China. This achieves a practical significance. However, the contribution and innovation points of this paper are not clear. The underlying comments can help the authors to improve further.

Main problems:

i.              The highlights section seems like the introduction to the research process, which should have briefly explained the contribution and innovation points of the article. In other words, you should remark on the new or significant findings.

ii.              In the introduction, the content of the first three paragraphs is not hard to understand, and the relationship between the contexts is relatively disordered.

iii.              For Table 1, the relationship between society and the government, residents, and enterprise can be briefly explained.

iv.              In Figure 1, the process drawn indicates that the estimates for direct costs and the estimates for emissions are conducted simultaneously, which is different from the order explained by the article.

v.              I doubt that the cost calculation does not involve the management cost of this part, and there is no special explanation.

vi.              In Equation 3, one letter V is missing.

vii.              I highly did not understand the time horizon of the data resource “from 2008 to 2015”. After all, that of 2008 to 2012 is far away from the current transportation trend. Whatever the reasons or limitations, the authors should explain more explicitly. Moreover, this part indeed is a bit contradictory to the first paragraph of the Introduction (whose data can trace back to 2021).

viii.              Re results, the article spent a lot of text explaining some obvious information in the picture, in fact, just needs to summarize the general meaning of the picture to the reader. Some comparisons between different regions lack substantial significance for the article.

ix.              In discussion, according to the disadvantages of the methods used in this paper, corresponding improvement measures should be considered.

x.              At the end of the conclusion, you should briefly mention the development prospects of the proposed method or clarify the feasibility of the employed method. What aspects are improved?

Minor problems:

xi.              In keywords, there is scope overlap between yellow-label vehicle and high-emission vehicle, and yellow-label vehicle only appears once in the text section. You should choose to keep the latter and delete the former.

xii.              Finally, it perhaps is not a minor issue. In my view, the language of the paper should have been checked and ensured before submission but perhaps not. Many inappropriate expressions impede the merits of the paper although I understand the authors are not native speakers.

Reviewer 2 Report

This work is based on the cost-benefit analysis (CBA) of policy restrictions for high-emission vehicles (HEV) in the Hebei province (China), including Beijing and many other densely-populated cities with a large number of HEVs.

In my opinion CBA is a valuable approach, needed to correctly quantify direct economic, societal and environmental impact costs.

The methodological approach is convincing. The authors include an up-to-date (i.e. WRF-CMAQ) air quality system to quantify the impact on the environment, and correctly outline a method to quantify the impact on human health and economic costs. I have no major concern.

Minor suggestions

1/ The authors refer to the bibliographical references n. 18-20 for the Pollution Prevention and Control Action Plan, but these references are written in Chinese and are not available to a wider audience. I suggest that you include an additional paragraph that summarizes the most important features of this plan, in order to present your results in plain manner.

Reviewer 3 Report

Dear authors!

Interesting theme and topics of the article. Your methodology is not precise defined, I could say it has a lot of shallow assumptions. While you made a lot of computations, it will be interesting to represent them in a more reader-friendly way. Please see MS Word file for comments and some questions you to improve the article.

Regards.

Round 2

Reviewer 1 Report

Thanks for the authors’ revised manuscript (ijerph-2031266). The authors addressed a majority of comments and the report was improved. However, there is still some format error to edit. Of course, they are not core but motivate the authors to check more. Besides, the language issues should have been edited.

Reviewer 3 Report

Dear Authors!

You explained some comments we asked for answers to, but still, I believe that you can make some t-tests to demonstrate statistically the improvements (significant yes or no). And at the end, it will be interesting to convert monetary units of benefits to human lives saved.

Best regards.
